# Hepatitis C Virus Treatment Status and Barriers among Patients in Methadone Maintenance Treatment Clinics in Guangdong Province, China: A Cross-Sectional, Observational Study

**DOI:** 10.3390/ijerph16224436

**Published:** 2019-11-12

**Authors:** Yin Liu, Xia Zou, Wen Chen, Cheng Gong, Li Ling

**Affiliations:** Department of Medical Statistics, School of Public Health, Sun Yat-sen University, Guangzhou 510080, China; liuy429@mail2.sysu.edu.cn (Y.L.); zouxia@mail3.sysu.edu.cn (X.Z.); chenw43@mail.sysu.edu.cn (W.C.); gongch25@mail2.sysu.edu.cn (C.G.)

**Keywords:** methadone maintenance treatment, hepatitis C virus, treatment experience, directly observed treatment service, direct-acting antivirals

## Abstract

We aimed to evaluate the status and barriers related to hepatitis C virus (HCV) treatment among Chinese methadone maintenance treatment (MMT) clients, and the willingness and barriers of patients to accept directly observed treatment (DOT) service and oral direct-acting antivirals (DAAs). We conducted a cross-sectional survey from July to October 2017 in Guangdong Province, China, involving 678 HCV antibody-positive MMT patients. If they reported being infected with HCV, then their HCV treatment experience, willingness to use DOT and DAAs, along with any barriers, were collected. Logistic regression analysis was used to identify the correlates of initiating HCV treatment. Among those reporting HCV infection (54%, 366/678), 39% (144/366) initiated treatment; however, 38% (55/144) interrupted and 55% (79/135) delayed treatment for 15 months. Seventy-five percent (273/366) and 53% (195/366) were willing to use DOT and DAAs, respectively. Unaffordable medical costs and insignificant symptoms were the major barriers to HCV treatment and accepting DOT or DAAs. The lack of a stable residence, being a woman, and having ever injected drugs were all associated with a low probability of initiating treatment (*p* < 0.05). This study highlights a limited uptake of HCV treatment among MMT patients, and a need to strengthen the popularity of DOT and DAAs and integrate them into Chinese MMT clinics.

## 1. Introduction

Hepatitis C virus (HCV) infection is associated with an excess risk of chronic hepatitis, cirrhosis, hepatocellular carcinoma, and liver transplantation [1]. People who use drugs (PWUD), especially people who inject drugs (PWID), are the primary high-risk population for HCV infection [2]. Globally, approximately 52% of PWID are HCV antibody positive [3]. In 2017, 2.553 million PWUD were registered in China [4], and a national report suggested that up to 33.4% of PWUD and 61.2% of PWID were HCV antibody positive [5].

The World Health Organization (WHO) put forward a 2030 target for the elimination of viral hepatitis and the provision of treatment for 80% of diagnosed HCV cases [6]. The methadone maintenance treatment (MMT) program, a community-based opioid substitution therapy, provides a good opportunity to deliver interventions or medical services for HCV-infected PWUD and PWID [7]. China has the largest MMT program worldwide and has served approximately 190,000 clients [8]. While MMT has been shown to reduce drug use and the incidences of HCV and injection-related risk behaviors [9,10], the HCV antibody prevalence among Chinese MMT clients is still high at 60.1% [11]; furthermore, little is known about the status of and barriers for HCV treatment.

There are several studies regarding HCV treatment status in developed countries that focus on areas such as treatment uptake, treatment delay, and treatment completion and adherence, but these studies are not suitable for China and other resource-limited countries due to differences in MMT programs and health care approaches. In China, only HCV education and HCV-antibody testing are provided to Chinese MMT clients; patients must go to specialized hospitals for further diagnosis and treatment for HCV while receiving MMT. However, in some developed countries directly observed treatment (DOT) service has been a successful method for the delivery of HCV treatment in MMT programs [12,13,14], in these services, HCV evaluation (including HCV antibody/RNA/genotype testing and liver function monitoring) and treatment services were given under supervision alongside addiction management in MMT clinics. That is to say, HCV patients can be evaluated and treated in MMT clinics instead of specialized hospitals, and medical costs, and payment methods remain unchanged. On the other hand, new oral direct-acting antivirals (DAAs) for HCV treatment have been fully implemented in many developed countries [6], while in China and many other resource-limited countries where DAAs are newly or not yet approved, a combined injection of peg-interferon-α with ribavirin (PR) is still the frontline HCV therapy [15]. Compared with PR, oral DAAs have an higher sustained virologic response (SVR) of 91–99%, fewer side effects and a shorter treatment course for 12–24 weeks [16]. Different therapies may lead to different rates of initiating treatment for HCV. Therefore, to achieve the WHO’s target, it is necessary to determine HCV treatment status of Chinese MMT patients along with any barriers they encountered. Previous studies in China have reported that few HCV-infected MMT clients had ever received treatment [17,18], but the characteristics of patients who were at high risk for not starting treatment were not identified. Additionally, regular and uninterrupted treatment and shorter times between diagnosis and treatment are associated with a higher SVR [19,20], but the proportion of treatment discontinuation, the reasons for discontinuing treatment and the delay between diagnosis and treatment were all unreported among HCV-infected individuals in Chinese MMT clinics.

Integrating DOT service into MMT programs and DAAs have been reported as facilitating access to HCV treatment [6,12,13,14]. Therefore, in order to promote the future implementation of DOT service into Chinese MMT programs, it is critical to understand patients’ willingness and the barriers they face in accepting the DOT program. Since July 2017, the first generic DAAs (daclatasvir plus asunaprevir) have been on the market on the Chinese mainland [21] and cost RMB 60,000 (approximately USD 8800) for a 24-week course [22]. In the early era of DAAs, the level of acceptance of these drugs was not been reported. To improve the availability of DAAs worldwide, it is necessary to explore the acceptance of and barriers to receiving DAAs as HCV therapy.

Therefore, with the aim of providing reference data for policy makers and health authorities to curb the HCV epidemic, this study was conducted to assess HCV treatment status, including the uptake of HCV treatment, the proportion of treatment discontinuation, the reasons for discontinuing treatment and the delay time between diagnosis and treatment. Additionally, we explored the characteristics of patients who were unlikely to initiate treatment, and the willingness to accept (and barriers against accepting) DOT and DAAs among HCV antibody positive MMT clients.

## 2. Materials and Methods

### 2.1. Design and Setting

This cross-sectional survey study was conducted in Guangdong province, China, from July to October 2017. Guangdong province is located in southern China and has the largest number (457,000) of documented PWUD in 2017, accounting for approximately 17.9% of all documented Chinese PWUD [23]. As of 2016, 63 MMT clinics in 18 cities had been established in Guangdong [24]. The prevalence of HCV antibody positive among MMT clients was estimated to be approximately 70% in Guangdong province [25,26]. Sixteen clinics were selected by a two-step stratified random sampling procedure. All MMT clinics in Guangdong province have the same model of HCV care, including HCV education and HCV antibody testing. All MMT clinics provide free HCV antibody testing and Human immune-deficiency virus (HIV)-1/2 testing for clients upon enrolment and every six months. HCV antibodies were tested using enzyme linked immunosorbent assay (ELISA) (Aibo Biotech Company, Hangzhou, China). For further diagnosis, clients whose HCV antibody is positive need to go to specialized hospitals for HCV RNA testing. Clinical doctors in the specialized hospitals would test HCV genotype to determine treatment strategy, because the treatment outcomes of PR were related with HCV genotypes [27].

Previous evidence showed that at the MMT level, the model of HCV care and geographic locations (rural or urban) were related to the uptake of HCV treatment [28,29,30]. Hence, the 63 MMT clinics were divided into two groups on their geographic locations: twenty-four and thirty-nine MMT clinics were located in rural and urban areas, respectively; then 25% clinics (six rural and ten urban) were selected randomly from each group.

### 2.2. Participants

All participants at the selected MMT clinics were included in this study if they: (1) agreed to share their HIV-1/2 and HCV antibody testing records at the MMT clinics; (2) provided informed consent. Participants were excluded if their HCV antibody tests had been negative within six months according to the testing records at the MMT clinics.

### 2.3. Data Collection

An interviewer administered a survey on the participants’ self-reported HCV infection, HCV treatment experience and barriers, and patients’ willingness to accept and barriers against accepting DOT and DAAs. Andersen’s behavior model for healthcare utilization was the framework used as a planning tool to help understand patients’ initiation or completion of HCV treatment through predisposing, enabling and need factors [31]. Factors considered for inclusion in the model were self-reported perceived barriers and patient characteristics associated with initiating HCV treatment (Figure 1).

This questionnaire took the participants approximately 15 min to complete; after completion, the participants received RMB 20 (approximately 3 USD) for their time. Referrals and consultations for HCV treatment were provided to all participants who reported an infection with HCV during the interview. After completing the interviews, we collected the participants’ latest (within six months) HIV and HCV testing records from the MMT clinics and linked their latest HCV testing records to the surveys using the treatment identification numbers for methadone used in the clinics. If a participant’s self-reported HCV infection was inconsistent with the test result, we would recommend that the MMT clinic inform the participant of the test result and provide referral services for HCV treatment.

### 2.4. Measures

#### 2.4.1. Self-Reported HCV Infection Status

The participants were asked, “Are you currently or have you ever been infected with HCV?” If they answered, “Yes, I am infected with HCV” or “Yes, I was infected with HCV, but now I am cured”, they were regarded as “Reporting infection”. If the participants answered, “No, I have never been infected with HCV”, they were regarded as “Reporting no infection”. If the participants answered, “I do not know whether I’ve been infected with HCV”, they were regarded as “Unknown”.

#### 2.4.2. HCV Treatment Status

If the participants were regarded as “Reporting infection”, information on when they became aware of the infection, current HCV treatment status (among four options: never treated, currently being treated, received treatment but did not complete the full course, completed treatment and was cured), and treatment sites and medications for HCV were collected. If they reported that they had never been treated for HCV, they were regarded as “Not initiating treatment”, and the related reasons were investigated in further interviews. If they reported that they were currently undergoing treatment, had ever received treatment but did not complete the course, or completed treatment and were cured, they were regarded as “Initiating treatment”, and the treatment start time, treatment sites and medication, and reasons for discontinuing treatment were investigated through further interviews.

#### 2.4.3. Patients Characteristics

Patient characteristics were collected based on Andersen’s behavior model; all characteristics were independent variables related to the initiation of treatment, although some factors were excluded due to expected collinearity. The patient characteristics were collected from the participants’ self-reports, except for the data on duration of MMT and HIV infection status, which were determined according to the records at the MMT clinics. Housing status was categorized as having a stable residence or not having stable residence (staying in a shelter, motel, or detoxification center, or staying with various acquaintances).

#### 2.4.4. Willingness to Accept DOT and Barriers Against It

For participants who were “Reporting infection”, after the DOT was introduced to them, then the following question was asked: “If HCV evaluation and medical treatment were implemented in MMT, but you still had to pay the medical costs out-of-pocket or through health insurance if you have it, would you use it and initiate HCV treatment?” If participants answered, “Yes, I think I would use it”, they were regarded as “being willing to use DOT”. If participants answered, “No, I would not use it”, the related reasons were investigated in further interviews.

#### 2.4.5. Willingness to Use DAAs and Barriers against Using Them

For participants “Reporting infection”, after DAAs were introduced to them, then the following question was asked: “New oral direct antivirals have a shorter treatment course lasting 12–24 weeks, few side effects and a cure rate of 91–99%, but if you had to pay approximately RMB 60,000 (approximately USD 8800) [22] for these medicines out of your own pocket, would you use them?”. If the participants answered, “Yes, I think I would use these antivirals”, they were regarded as “being willing to use DAAs”. If the participants answered, “No, I would not use them”, the related reasons were investigated in further interviews.

### 2.5. Statistical Analysis

A database was constructed using Epidata V3.0 with double entry. All statistical analyses were performed with SAS V9.4. The following were calculated: (1) the proportion of patients who reported being or having ever been infected out of all HCV-antibody positive clients; (2) the proportion of patients initiating treatment for HCV among those reporting HCV infection; (3) the proportions of patients who were currently undergoing treatment, had discontinued treatment, and had completed treatment and were cured among those initiating treatment; and (4) the delay between becoming aware of the infection and initiating treatment.

Descriptive analyses were used to report barriers and characteristics based on Andersen’s behavior model. Means and standard deviations were used for normally distributed continuous variables, and medians and inter-quartile ranges (IQR) were used for non-normally distributed continuous variables. Categorical variables were described as frequency distributions and percentages.

Bivariable and multivariable logistic regressions were performed to assess the association of each potential factor with initiating treatment among those reporting HCV infection. All significant bivariable predictors (*p* < 0.10) were selected to be adjusted for multivariable logistic regression, and only those variables with a *p*-value < 0.05 were retained in the final multivariable model. Crude odds ratios (COR) with 95% confidence intervals (CI) were maintained for bivariable logistic regressions, and adjusted odds ratios (AOR) with 95% CIs were reported for multivariable logistic regressions.

## 3. Results

### 3.1. Patient Characteristics According to Self-Reported HCV Status

A total of 858 clients were interviewed, after excluding those whose HCV antibody tests had been negative within six months according to the testing records at the MMT clinics, finally 678 HCV antibody-positive patients were included in this study. Among them, 366 (54%) reported that they were currently or had previously been infected with HCV (Reporting infection), 259 (38%) reported they had never been infected with HCV (Reporting no infection), and 53 (8%) did not know their infection status (Unknown).

Patient characteristics according to reported HCV status are shown in Appendix A. In general, most of the HCV antibody positive patients were male (88%), married (61%), covered by health insurance (73%), employed (57%); had a stable residence (87%); and had not drunk alcohol in the past month (69%). The mean age was 43.3 ± 6.65 years, with an average duration of drug abuse prior to enrolling in MMT of 14.3 ± 5.95 years. Eighty-one percent of the patients were PWID, and 11% and 6% were co-infected with HIV and HBV.

### 3.2. HCV Treatment Status among Those in the MMT Program who Reported HCV Infection

Of those who reported HCV infection, 39% (144/366) started antiviral treatment for HCV (Initiating treatment), and 61% (222/366) were never treated (Not initiating treatment). Of those initiating treatment, 32% (46/144) were currently undergoing treatment, 38% (55/144) did not complete the full course of treatment, and 30% (43/144) had completed treatment and were cured.

Of those who initiated treatment, 139 were treated with an interferon-based treatment at a general hospital or a specialized hospital for infectious diseases, while another five patients were treated using DAAs through medical tourism in India and they were all cured.

After becoming aware of their infection, 55% (79/144) of the participants delayed treatment for a median of 15 months (IQR:44, 3–47). Specifically, among patients who were currently undergoing treatment, did not complete the full course of treatment, or completed treatment and were cured, 52% (25/46) delayed treatment for a median of 11 (IQR:83, 6–89) months, 64% (35/55) delayed treatment for a median of 15 (IQR:45, 3–48) months, and 44% (19/43) delayed treatment for a median of 15 (IQR:43, 3–46) months, respectively.

### 3.3. Reasons for Not Initiating Treatment among Those in the MMT Program Who Reported HCV Infection

The reasons for not initiating treatment are shown in Figure 2. The most common reasons included unaffordable medical costs (40%, 89/222), not needing treatment due to mild symptoms (33%, 74/222), not knowing where to get treated (19%, 43/222), and not being convinced that the treatment works (12%, 26/222).

### 3.4. Reasons for Discontinuing Treatment among Those Initiating Treatment for HCV in the MMT Program

The reasons for discontinuing treatment are shown in Figure 3. The most common reasons were no need for treatment due to improved health status (24%, 13/55) and unaffordable medical costs (22%, 12/55). In addition, long distances between hospitals and MMT clinics (18%, 10/55), long course of treatment (18%, 10/55), poor virologic responses and side effects of treatment (16%, 9/55) were also common reasons.

### 3.5. Factors for Not Initiating Treatment among Those in the MMT Program Reporting HCV Infection

Factors related to initiating treatment are shown in Table 1. Multivariable logistic regression analyses indicated that being a woman (AOR: 0.40, 95% CI: 0.18–0.86), having ever injected drugs (AOR: 0.44, 95% CI: 0.24–0.81) and not having a stable residence (AOR: 0.26, 95% CI: 0.25–0.99) were factors in individuals who were less likely to access any treatment, while those who were co-infected with HIV (AOR: 2.83, 95% CI: 1.50–5.33) and HBV (AOR: 2.46, 95% CI: 1.12–5.39) were more likely to initiate treatment for HCV.

### 3.6. Willingness to Use DOT in MMT Clinics among Those in the MMT Program Reporting HCV Infection

After those reporting HCV infections were informed about DOT, 75% (273/366) expressed willingness to use DOT and initiate treatment. Specifically, among those who did and did not initiate treatment, 90% (130/144) and 64% (143/222) expressed willingness to accept DOT, respectively. The reasons patients gave for being unwilling to accept DOT are shown in Figure 4. Unaffordable medical costs (72%, 67/93), no need due to mild symptoms (16%, 15/93), and no time to be treated frequently (11%, 10/93) were the major barriers to accepting DOT. Additionally, not being convinced that treatment works (6%, 6/93) and fear of treatment side effects (6%, 6/93) were also reasons given for dismissing DOT.

### 3.7. Willingness to use DAAs in MMT Clinics among Those in the MMT Program Reporting HCV Infection

After DAAs were introduced to those who reported HCV infection, 53% (195/366) expressed a willingness to use DAAs and initiate treatment. Specifically, among those who did and did not initiate treatment, 73% (105/144) and 40% (90/222) expressed a willingness to accept DAAs, respectively. The reasons for patients’ unwillingness to accept DAAs are shown in Figure 5. Unaffordable medical costs (74%, 127/171) and no need for treatment due to very mild symptoms (10%, 18/171) were the major barriers to accepting DAAs. Not believing in the effectiveness of DAAs (9%, 15/171) was another common reason.

## 4. Discussion

This study uncovered important issues among Chinese MMT clients who need more attention, including high rates of HCV infection (79%) and a low proportion of HCV treatment initiation (21%) among all HCV-infected patients and a high rate of treatment discontinuation (38%) and frequent delays of treatment (55%) among those who initiate treatment. Unaffordable medical costs and self-reported mild symptoms were the most frequently reported barriers stopping patients from accessing or completing HCV treatment. Although the patients reported a willingness to accept DOT (75%) and DAAs (53%), these two factors may also prevent them from receiving treatment.

The rate of HCV infection reported in this study was higher but the uptake of HCV treatment was lower than reported for other developed countries, especially those that integrate comprehensive HCV care into MMT [12,13,14]. The low proportion of patients reporting HCV infection is the major reason for this difference. In our study, 46% of HCV antibody positive individuals reported that they had never been infected with HCV or did not know their infection status. On the one hand, the patients may conceal their infection status for fear of discrimination, or they may neglect their HCV infection due to limited HCV knowledge [17]. On the other hand, MMT staff members may neglect to inform clients about their HCV testing results due to their own limited HCV-related knowledge [17,18]. Additionally, referral services for HCV are not mandatory in Chinese MMT programs [32], which may also reduce the staff’s determination to inform patients about their HCV testing results. In our survey, we found that some staff had gradually stopped informing clients about infections because most clients would not initiate treatment despite knowing they were infected. Thus, it is necessary to strengthen publicity and education regarding HCV among MMT staff members and clients, and help reduce stigma and discrimination.

Unaffordable medical costs were the predominant barriers against initiating or completing treatment and accepting DAAs. More than half of MMT clients have monthly incomes less than RMB 3000 (approximately USD 440), which covers only daily spending needs in Guangdong province. In China, the current, standard 48-week price for PR is approximately RMB 46,447 (approximately USD 6835). While PR is listed in the medical insurance directory, and over 70% of clients have medical insurance, the reimbursement rate for PR is limited, and the costs of treatment during hospitalization also have set limits. By 2019, many other DAAs, such as ombitasvir/paritaprevir/ritonavir+dasabuvir for genotype 1b and sofosbuvir+velpatasvir/daclatasvir for all genotypes, had been approved in China. The efficiency and costs of these regimens were similar to DCV + ASV [33]. Previous studies have proven that DAAs were highly cost-effective compared with PR [34,35], however, as DAAs were not included in the national health insurance drug lists, patients had to pay out of their own pockets [36].

Mild symptoms of early HCV was another major reason for not initiating or completing treatment. In despite of the high SVR of DAAs, it was also the major barrier for accepting DAAs. According to a previous study conducted among HCV patients in MMT clinics in Yunnan and Gansu provinces, approximately 85% of patients are chronically infected with HCV without cirrhosis [37]. We also collected information about the stage of hepatitis through patients’ self-reports, but 44% (160/366) of patients did not know their stage; further, only 5% (10/206) of those who knew their stage reported having cirrhosis. As the symptoms of early fibrosis are not obvious [38,39], and patients were not aware of the severity of their HCV infection [17,18], they did not want to pay for early medical care or to complete treatment. In developed countries, all chronic HCV patients were recommended to initiate treatment regardless of fibrosis stage, and early treatment was more cost-effective, for both DAA and PR regimens [40]. However, evidence on cost-effectiveness of early treatment for chronic HCV patients was limited among resource-limited settings. Only one cost-effectiveness analysis modelled in Egypt suggested that immediate treatment of patients with early fibrosis stage was less expensive and more effective than using DAAs in the general population [41], but no studies were conducted among PWID in resource-limited settings. PWID are at increased risk of HCV transmission and reinfection [38] and HCV infection progresses slowly in most patients. Due to the high price of treatment, further cost-effectiveness analyses are also needed to determine whether early-fibrosis stage patients should be treated immediately in China and other resource-limited settings. Anyway, to avoid liver cirrhosis or hepatocellular carcinoma, liver function monitoring should be provided to patients who did not initiate or complete treatment [15]. Therefore, it may be time to implement DOT services in the MMT program to facilitate access to early diagnosis and improve patients’ awareness of their fibrosis staging. Chinese MMT patients also showed a strong willingness to accept DOT. Even in the ‘Not initiating treatment’ group, 64% of patients were ready to accept DOT and initiate treatment, suggesting that integrating HCV treatment into MMT would also accelerate the uptake of HCV treatment. Further studies are needed to explore the cost-effectiveness of DOT if it were to be integrated into the Chinese MMT program.

Consistent with studies in developed countries [42,43], we identified that female patients, those without a stable residence, and those who had ever injected drugs were less likely to initiate HCV treatment. Female PWUD may encounter more discrimination and find it more difficult to receive support from their family and society than male PWUD [44]. Housing status is correlated with social support or economic status [45]. Patients with a stable residence showed more willingness to complete regular antiviral treatments than those who did not know where they would stay during the treatment period [46]. PWID often encounter more significant barriers to accessing HCV treatment and are often denied treatment because of concerns regarding on-going risky behaviors and reinfection following treatment [38,47,48]; these barriers are present despite recommendations that PWID be prioritized for treatment in resource-limited countries as they are the group at the greatest risk of HCV infection [16]. Thus, more attention and care should be paid to these subgroups and attempts to treat them fairly should be prioritized. 

In contrast to studies in some developed countries [49,50], in this study, MMT patients co-infected with HIV or HBV were more likely to initiate HCV treatment. Two reasons may account for this difference. First, those co-infected with HIV or HBV tend to have more serious symptoms of liver disease [51,52], which may receive more attention from patients and physicians and may lead to a greater likelihood of treatment initiation. Second, Chinese MMT clinics were originally started with the initiation of curbing HIV among drug users [53]. According to the current Chinese MMT instruction manual [32], MMT staff must refer HIV/AIDS patients to designated medical institutions for antiretroviral therapy. More attention and referrals for HIV patients may also increase the chance that people who are co-infected with HIV and HCV will be treated for HCV. However, dual infection treatment costs more, and the need for HCV treatment is almost always downplayed among those co-infected with HIV [54], which may result in high rates of HCV treatment interruption. In this study, approximately 52% (27/52) of dual-infected patients who initiated treatment did not complete it. In this respect, it is crucial to promote interventions to prevent co-infection, such as providing sterile injection equipment, promoting the consistent use of condoms, immunization with hepatitis B vaccines, and so on. For patients who have already been co-infected, DAA-based regimens should be prioritized as they have an SVR of 91–99%, even among those co-infected with HIV– or HBV–HCV [55].

The study has several limitations. First, compared with other provinces of China, Guangdong province is well-developed and provides more jobs. Uptake of HCV treatment might be overestimated when it was applied to the all Chinese MMT patients. Second, the self-reported data may be subject to social desirability and recall biases. Third, because the report relied on a cross-sectional survey design, causal relationships between patient characteristics and treatment initiation could not be firmly established. Finally, due to the fact approximately 26% of the acute-infected patients would spontaneously clear infection without any treatment [56], we could not determine the number of patients who would be eligible for treatment because of the lack of access to HCV RNA testing according to current Chinese guidelines, which suggest that the patient must be positive for HCV RNA before treatment is initiated [15]. However, given that drug abusers are frequently exposed to risk factors for HCV infection, it is assumed that a large proportion would meet the eligibility criteria [38,57].

## 5. Conclusions

This study uncovered a high prevalence of HCV antibody positive patients, but it also showed limited treatment uptake, high rates of treatment discontinuation, and frequent delays in treatment among HCV antibody positive patients in Chinese MMT clinics. Patients exhibited a strong willingness to use DOT and DAAs but encountered barriers, such as unaffordable fees and a lack of HCV-related knowledge. We recommend the implementation of comprehensive HCV care with high treatment reimbursement in the Chinese MMT program to increase patient access to treatment.

## Figures and Tables

**Figure 1 ijerph-16-04436-f001:**
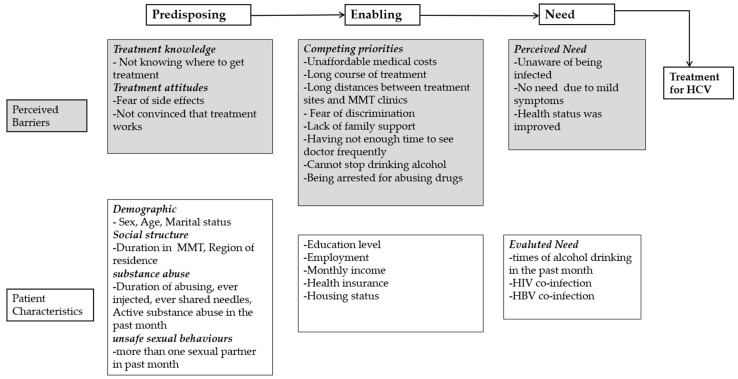
Framework of predisposing factors, enabling factors and the need for hepatitis C virus (HCV) treatment. Methadone maintenance treatment (MMT). Human immune-deficiency virus (HIV). Hepatitis B virus (HBV)

**Figure 2 ijerph-16-04436-f002:**
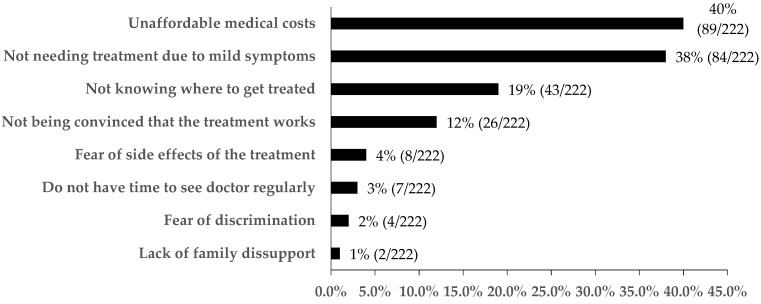
Reasons for not initiating treatment among patients who reported HCV infection, Guangdong, China (n = 222).

**Figure 3 ijerph-16-04436-f003:**
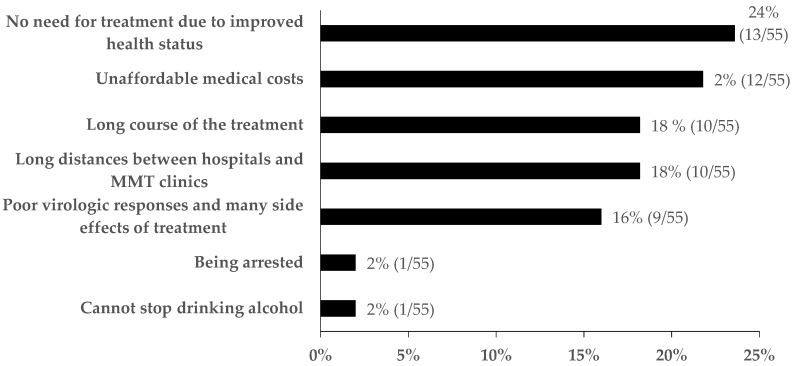
Reasons for discontinuing treatment among patients who had initiated treatment for HCV, Guangdong, China (n = 55).

**Figure 4 ijerph-16-04436-f004:**
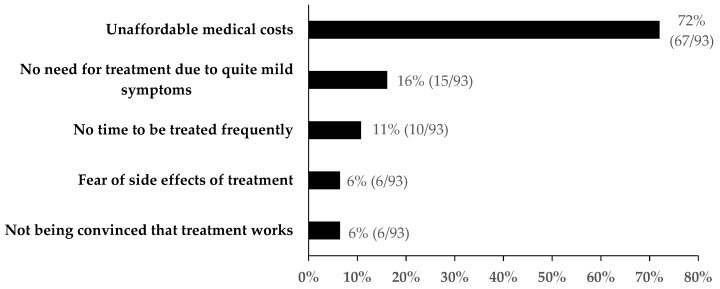
Reasons for dismissing directly observed treatment (DOT) among patients who reported HCV infection, Guangdong, China (n = 93).

**Figure 5 ijerph-16-04436-f005:**
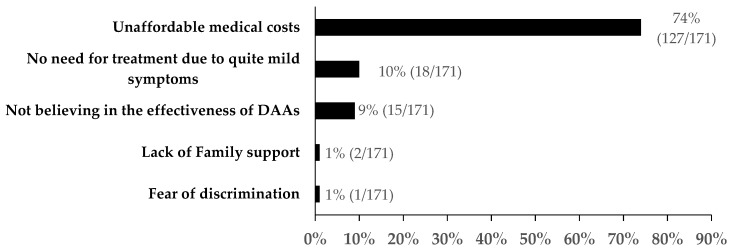
Reasons for dismissing direct-acting antivirals (DAAs) among patients who reported HCV infection, Guangdong, China (n = 171).

**Table 1 ijerph-16-04436-t001:** HCV treatment status by patient characteristics among patients who reported HCV infection, Guangdong, China (n = 366).

Variables	Initiating Treatment (%)		COR (95% CI)	*p*	AOR (95% CI)	*p*
	No (n = 222)	Yes (n = 144)				
***Predisposing factors***						
**Gender**				0.016		0.019
male	188 (58)	134 (42)	Reference		Reference	
female	34 (77)	10 (23)	0.41 (0.20–0.86)		0.40 (0.18–0.86)	
**Age**				0.663		
18–34	20 (63)	12 (37)	Reference			
34–44	126 (62)	76 (38)	1.01 (0.47–2.17)			
≥45	76 (58)	56 (42.)	1.23 (0.56–2.72)			
$\bar{x}\pm s$	42.5 ± 6.25	42.9 ± 6.08	1.01 (0.98–1.05)			
**Marital Status**				0.278		
Married	128 (58)	91 (42)	Reference			
Single	58 (60)	38 (40)	0.92 (0.57–1.50)			
Divorced or windowed	36 (71)	15 (29)	0.59 (0.30–1.13)			
**Duration in MMT (years)**				0.083		0.173
<5	64 (54)	54 (46)	Reference		Reference	
≥5	158 (64)	90 (36)	0.68 (0.43–1.05)		0.72 (0.45–1.16)	
**Region of residence**				0.994		
urban	151 (61)	98 (39)	Reference			
rural	71 (61)	46 (39)	1.00 (0.64, 1.57)			
**Duration of abusing drugs(years) ^#^**				0.369		
<10	44 (59)	31 (41)	Reference			
10–19	144 (63)	86 (37)	0.85 (0.50–1.44)			
≥20	30 (53)	27 (47)	1.28 (0.64–2.56)			
$\bar{x}\pm s$	14.0 ± 5.47	14.3 ± 6.03	0.99 (0.96–1.03)			
**Ever injected drugs**				0.048		0.008
No	29 (49)	30 (51)	Reference		Reference	
Yes	193 (63)	114 (37)	0.57 (0.33–1.00)		0.44 (0.24–0.81)	
**Ever shared needles**				0.336		
No	160 (62)	97 (38)	Reference			
Yes	62 (57)	47 (43)	1.25 (0.79–1.97)			
**Abusing drugs in the past month**				0.397		
No	194 (56)	130 (40)	Reference			
Yes	28 (67)	14 (33)	0.75 (0.38–1.47)			
**More than one sexual partner in the past month**				0.343		
No	213 (61)	135 (39)	Reference			
Yes	9 (50)	9 (50)	1.58 (0.61–4.07)			
***Enabling factors***						
**Education level**				0.386		
≤Primary school	56 (66)	29 (34)	Reference			
Junior high school	124 (61)	81 (39)	1.26 (0.74–2.14)			
≥Senior high school	42 (55)	34 (45)	1.56 (0.83–2.96)			
**Employment**				0.618		
Unemployed	102 (59)	70 (41)	Reference			
Employed	120 (62)	74 (38)	0.90 (0.59–1.37)			
**Monthly income**				0.436		
<3000	132 (61)	85 (39)	Reference			
3000–5000	71 (64)	40 (36)	0.88 (0.55–1.41)			
≥5000	17 (51)	16 (49)	1.46 (0.70–3.05)			
**Have health insurance**				0.616		
Yes	163 (60)	109 (40)	Reference			
No	56 (63)	33 (37)	0.88 (0.54–1.44)			
**Have stable residence**				0.074		0.049
Yes	181 (59)	127 (41)	Reference		Reference	
No	40 (71)	16 (29)	0.57 (0.31–1.06)		0.26 (0.25–0.99)	
***Need factors***						
**Drinking times in the past month**				0.484		
Never	160 (62)	98 (38)	Reference			
1–3 times per month	36 (63)	21 (37)	0.95 (0.52–1.73)			
≥1 times per week	13 (52)	12 (48)	1.51 (0.66–3.44)			
≥1 times per day	13 (52)	12 (48)	1.51 (0.66–3.44)			
**HIV**				0.001		0.001
No	201 (64)	112 (36)	Reference		Reference	
Yes	21 (40)	32 (60)	2.73 (1.51–4.97)		2.83 (1.50–5.33)	
**HBV**				0.015		0.024
No	209 (63)	125 (37)	Reference		Reference	
Yes	13 (41)	19 (59)	2.44 (1.17–5.12)		2.46 (1.12–5.39)	

*CI*, confidence interval. *COR*, crude odds ratio. *AOR*, adjusted odds ratio. # before entering in MMT.

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
