# Peer review of "Hepatitis C Virus Treatment Status and Barriers among Patients in Methadone Maintenance Treatment Clinics in Guangdong Province, China: A Cross-Sectional, Observational Study"

_ijerph, 2019, doi:10.3390/ijerph16224436_

Round 1

Reviewer 1 Report

The authors have tried to address the criticisms of the initial reviewer but were not able to satisfy that reviewer.   The authors should reconsider the lack of cost effectivenenss data included in the ms as the reviewer proposed.   It is a good idea, although its absence does not detract from the quality of the current presentation 

Author Response

Response to Reviewer 1 Comments

Comments and Suggestions for Authors:The authors have tried to address the criticisms of the initial reviewer but were not able to satisfy that reviewer.   The authors should reconsider the lack of cost effectiveness data included in the ms as the reviewer proposed.   It is a good idea, although its absence does not detract from the quality of the current presentation 

Response:Thanks for the reviewer’s suggestion. we acknowledged that the cost-effectiveness of treating chronic patients at different fibrosis stage was a good idea, and we have added a sentence in Discussion section to state “Further analyses are also needed to determine the cost-effectiveness of early treatment for chronic HCV patients in China and other resource-limited settings.” In fact, we have already conducted this cost-effectiveness analysis in another paper (which has been under review), and we found early treatment was cost-effective among patients who injected drugs, for both IFN and DAAs regimen.

Reviewer 2 Report

Liu et al. have done a cross-sectional, observational study on methadone maintenance clinic (MMT) patients' hepatitis c treatment status and barriers in the Guangdong province in China.

The authors conclude that there’s limited uptake of HCV treatment among MMT patients, and there is a strengthen the popularity of the directly observed treatment (DOT) and oral direct-acting antivirals (DAA’s) into the Chinese MMT clinics.

Here are my comments,

1.Page 2. Paragraph 2, Line 59.

What do the authors mean by ‘Different HCV therapy might be related to initiating treatment’.?

The use of the word Anti-HCV in the manuscript is confusing. Can use Hep C positive or Hep C ab +ve.

It is observed throughout the manuscript.

E.g., The anti- HCV prevalence among MMT clients was estimated to be approximately 70% in Guangdong province.

3.Page 2. Under the Design and settings: Can include the way the Hep C testing is done, including the details on the test and details on the genotypes as they paly role in predicting the outcomes of treatment.

Page 4.

2.4.4.  I am not sure how the question asked by the authors, makes to conclude whether the patients are willing to use or not use DOT.

As DOT means, a method of drug administration in which a health care professional watches as a person takes each dose of a medication.

Also, the patients might be discouraged if asked, ‘you still had to pay for the service out-of-pocket or through health insurance if you have it, would you use it and initiate HCV treatment?

I need clarification from the authors.

Under discussion:

Page 10. Last paragraph, the last three lines, the authors jump to say that ‘DOT services in the MMT program to facilitate access to early diagnosis and improve patients’ awareness of their fibrosis staging.’

What do they mean?

At this point, I felt that it would be better for the authors to define DOT in the introduction in a short paragraph.

Author Response

Response to Reviewer 2 Comments

Comments and Suggestions for Authors: Liu et al. have done a cross-sectional, observational study on methadone maintenance clinic (MMT) patients' hepatitis c treatment status and barriers in the Guangdong province in China. The authors conclude that there’s limited uptake of HCV treatment among MMT patients, and there is a strengthen the popularity of the directly observed treatment (DOT) and oral direct-acting antivirals (DAA’s) into the Chinese MMT clinics.

 Here are my comments,

Comment 1. Page 2. Paragraph 2, Line 59. What do the authors mean by ‘Different HCV therapy might be related to initiating treatment’.?

Response: Sorry for this confusion. It means that different therapies (PR or DAAs) may lead to different rates of initiating treatment for HCV. We have revised this sentence.  

Comment 2. The use of the word Anti-HCV in the manuscript is confusing. Can use Hep C positive or Hep C ab +ve. It is observed throughout the manuscript. E.g., The anti- HCV prevalence among MMT clients was estimated to be approximately 70% in Guangdong province.

 Response: Thanks for the reviewer’s suggestion. We have revised.

Comment 3. Page 2. Under the Design and settings: Can include the way the Hep C testing is done, including the details on the test and details on the genotypes as they paly role in predicting the outcomes of treatment.

 Response: Thanks for the reviewer’s comments. We have included detailed information on HCV testing, treatment and genotypes. (See page 3 line 99-103)

Comment 4.  I am not sure how the question asked by the authors, makes to conclude whether the patients are willing to use or not use DOT.As DOT means, a method of drug administration in which a health care professional watches as a person takes each dose of a medication. Also, the patients might be discouraged if asked, ‘you still had to pay for the service out-of-pocket or through health insurance if you have it, would you use it and initiate HCV treatment?

I need clarification from the authors.

Response: Firstly, we gave introduction of DOT to the patients as follows: DOT does not mean that patients can receive HCV evaluation and treatment services for free, it just means that patients can be evaluated and treated in MMT clinics without the need to go to the specialized hospitals. Medical costs and payment methods for HCV remain unchanged. If patients have health insurance, they could reimburse the HCV medical costs; otherwise they need pay out-of-pocket. Then, we asked patients “If HCV evaluation and medical treatment were implemented in MMT, but you still had to pay the medical costs out-of-pocket or through health insurance if you have it, would you use DOT and initiate HCV treatment?”  

Comments 5. Under discussion: Page 10. Last paragraph, the last three lines, the authors jump to say that ‘DOT services in the MMT program to facilitate access to early diagnosis and improve patients’ awareness of their fibrosis staging.’ What do they mean? At this point, I felt that it would be better for the authors to define DOT in the introduction in a short paragraph.

 Response: Thanks for your comments. We have revised and added a detailed description of DOT in the introduction (See page 2 line 53-57).

Round 2

Reviewer 2 Report

The authors have modified the manuscript and addressed all my comments and suggestions.

This manuscript is a resubmission of an earlier submission. The following is a list of the peer review reports and author responses from that submission.

Round 1

Reviewer 1 Report

Summary: the manuscript entitled "Hepatitis C virus treatment status and barriers among 2 patients in methadone maintenance treatment clinics 3 in Guangdong province, China: a cross-sectional, observational study" evaluated the status and barriers related to different hepatitis C virus (HCV) treatments, including directly observed treatment (DOT) and oral direct-acting antivirals (DAAs). The authors surveyed 678 HCV antibody-positive MMT patients in Guangdong providence from July to Oct in 2017. The survey indicated around half (54%) of the people who have the HCV infection, only 39% initiated the treatment. In addition, for more than half of the patients who initiated this process, 38% and 55% of the treatments were interrupted or delayed respectively for various reasons. The survey also indicates the female patients who have no stable residence and used drugs are less likely to initiate the treatment. All these interesting conclusions partially reflected the current status of the HCV treatment and some social problems in China, which could potentially draw the attention of the whole society to HCV infection and treatment issue in China.

Major changes:

In general, the authors use many tables to present the data, which is not a direct and easy way for the audience to visualize the data and figures. For the patient characteristics, the author probably can move it to the supplemental data. For some other figures, such as fig2A, the author probably can use pie charts or bar plots. It will be better to present all the figures in a more visualized way. The author mentioned several ways for the HCV treatment, but did not indicate that which treatment has better cure rate. The authors can design more questions related to the effects of the different treatments, which could potentially give the audience of this study more guidance to choose the suitable treatment. Line233: The authors indicated for women, if they had no stable residence and used drugs, they were less likely to have the HCV treatment. However, the table 3 showed there are only 44 females in total are reported as HCV infection cases. The number is much smaller than male cases. I am not sure if these 44 cases are enough to represent the whole female population.

Minor changes:

Line24 “Female, those having no stable residence, patients who injected drugs 24 had a low probability of initiating treatment (P<0.05).”: there are some grammer mistakes in this sentence. Line 244 “After those reporting HCV infection were”: should be “infections”. 

Author Response

Response to Reviewer 1 Comments

Summary: the manuscript entitled "Hepatitis C virus treatment status and barriers among patients in methadone maintenance treatment clinics in Guangdong province, China: a cross-sectional, observational study" evaluated the status and barriers related to different hepatitis C virus (HCV) treatments, including directly observed treatment (DOT) and oral direct-acting antivirals (DAAs). The authors surveyed 678 HCV antibody-positive MMT patients in Guangdong providence from July to Oct in 2017. The survey indicated around half (54%) of the people who have the HCV infection, only 39% initiated the treatment. In addition, for more than half of the patients who initiated this process, 38% and 55% of the treatments were interrupted or delayed respectively for various reasons. The survey also indicates the female patients who have no stable residence and used drugs are less likely to initiate the treatment. All these interesting conclusions partially reflected the current status of the HCV treatment and some social problems in China, which could potentially draw the attention of the whole society to HCV infection and treatment issue in China.

Major changes:

Point 1: In general, the authors use many tables to present the data, which is not a direct and easy way for the audience to visualize the data and figures. For the patient characteristics, the author probably can move it to the supplemental data. For some other figures, such as fig2A, the author probably can use pie charts or bar plots. It will be better to present all the figures in a more visualized way.

Response 1: Thanks for the reviewer’s suggestion. We have moved the patient characteristics to the supplemental data, and presented Fig2A-Fig3B as bar charts.

Point 2: The author mentioned several ways for the HCV treatment, but did not indicate that which treatment has better cure rate. The authors can design more questions related to the effects of the different treatments, which could potentially give the audience of this study more guidance to choose the suitable treatment.

Response 2: Thanks for the reviewer’s suggestion. There are several ways for the HCV antiviral treatment nowadays, but only DCV+ASV and PR were on the market during our survey. As indicated in the manuscript, DCV+ASV have better cure rate than PR. Since 2018, more DAAs have been on the market. To give more guidance to choose the suitable treatment, we have added the effects of the different DAAs treatments for different genotypes in Discussion section as “The efficiency and costs of other DAAs were similar to DCV+ASV, and their treatment course was shorter for 12 weeks” (see Line 306-310).

Point 3: Line233: The authors indicated for women, if they had no stable residence and used drugs, they were less likely to have the HCV treatment. However, the table 3 showed there are only 44 females in total are reported as HCV infection cases. The number is much smaller than male cases. I am not sure if these 44 cases are enough to represent the whole female population.

Response 3: Thanks for the reviewer’s suggestion. Based on previously published literatures (Wang C, et al and Liu Y, et al), most Chinese MMT clients were men and only 6.8%-15% were female. This proportion is similar to that in this study. We obtained the participants using stratified random sampling and we estimated a sample of 365 (with 25~55 females) reporting HCV infections would be enough to estimate the total population based on the following formula.

 ;

α: Type I error, 0.05;

δ: Permissible error, 5%;

P: Prevalence of HCV treatment among those who reported HCV infection, 29%.

Thus, we believe 44 cases could represent the whole female population in Guangdong Province.

Reference:

1.Wang C, Shi CX, Rou K, et al. Baseline HCV antibody prevalence and risk factors among Drug users in China's National Methadone Maintenance Treatment Program. PLoS One 2016;11:e0147922.

Liu Y, Liu Y, Zou X, et al. Trends and factors in human immunodeficiency virus and/or hepatitis C virus testing and infection among injection drug users newly entering methadone maintenance treatment in Guangdong Province, China 2006-2013: a consecutive cross sectional study. BMJ Open. 2017, 7(7):e015524.

Minor changes:

Line24 “Female, those having no stable residence, patients who injected drugs 24 had a low probability of initiating treatment (P<0.05).”: there are some grammer mistakes in this  sentence. Line 244 “After those reporting HCV infection were”: should be “infections”. 

Response 4: Thanks for the reviewer’s suggestion. We have carefully done English grammar checking in the revised version of the manuscript and corrected the mistakes in grammar based on the reviewer’s suggestion.

Reviewer 2 Report

This study presents a large cross-sectional study on HCV status regarding treatment among Methadone maintenance users in a chinese province. The study is well presented, methods and statistics look like correct and conclusions are fair. One major weakness regarding this study is that treatment offered to patients is mostly based on IFN regimens, which in my opinion has no role nowadays. In this sense, opinions of patients in the study are highly influenced by therapeutic regimen and conclusions do not apply to most developed countries where DAA regimens are exclusively used. I would point out other points that in my opinion could improve/reinforce strategic or practical messages.

Only about 20% of HCV patients develop cirrhosis. It would be very interesting to know exactly the fibrosis stage of the patients to even reinforce or adjust accessibility to antiviral therapy. In eastern countries, the implementation of DAA therapies was gradually applied: first in most severely ill patients (including decompensated patients) with a later expansion to more stable (mild-moderate fibrosis) patients (eradication among population). This strategy has been shown to be effective and the most severe patients obtain initially the most benefit, liver function and portal hypertension improve and even most of them avoid the need for transplantation; HCC remains a concern to be screened. Linking with previous point, a step-by-step approach to eradicate HCV infection may contemplate the inclusion of stratifying techniques (Fibroscan? serologic fibrosis tests?) that allowed selecting the best candidates for therapies. These therapies should only include in my opinion DAA regimens. In a resource-limited environment, it can be applied with a gradual intervention that prioritizes the sickest patients based on these techniques. In the meanwhile, more stable patients wait for DAA costs reduction or a universal access to DAA. We must remember that HCV infection progresses along years, slowly in most patients, so there’s place to be compliant. This paragraph and the previous one must be included in Discussion section since adding relevant information not initially taken into account. Authors should present an approximate prevalence of HCV infection in their population based on their data. With this they could really estimate an impact of HCV patients among Methadone users and a potential improvement by applying eradicating therapies. Linking with previous: Based on DAA potential costs, estimated prevalence and the costs from complications of advanced HCV patients (cirrhotic), authors could attempt to explore a costs-effects analysis (Markov model or similar) to propose intervention strategies based on their local data and resources. Introduction section must be shortened. Discussion should be shortened/modified according to my previous suggestions.

Author Response

Response to Reviewer 2 Comments

This study presents a large cross-sectional study on HCV status regarding treatment among Methadone maintenance users in a Chinese province. The study is well presented, methods and statistics look like correct and conclusions are fair. One major weakness regarding this study is that treatment offered to patients is mostly based on IFN regimens, which in my opinion has no role nowadays. In this sense, opinions of patients in the study are highly influenced by therapeutic regimen and conclusions do not apply to most developed countries where DAA regimens are exclusively used. I would point out other points that in my opinion could improve/reinforce strategic or practical messages.

Only about 20% of HCV patients develop cirrhosis. It would be very interesting to know exactly the fibrosis stage of the patients to even reinforce or adjust accessibility to antiviral therapy. In eastern countries, the implementation of DAA therapies was gradually applied: first in most severely ill patients (including decompensated patients) with a later expansion to more stable (mild-moderate fibrosis) patients (eradication among population). This strategy has been shown to be effective and the most severe patients obtain initially the most benefit, liver function and portal hypertension improve and even most of them avoid the need for transplantation; HCC remains a concern to be screened. Linking with previous point, a step-by-step approach to eradicate HCV infection may contemplate the inclusion of stratifying techniques (Fibroscan? serologic fibrosis tests?) that allowed selecting the best candidates for therapies. These therapies should only include in my opinion DAA regimens. In a resource-limited environment, it can be applied with a gradual intervention that prioritizes the sickest patients based on these techniques. In the meanwhile, more stable patients wait for DAA costs reduction or a universal access to DAA. We must remember that HCV infection progresses along years, slowly in most patients, so there’s place to be compliant. This paragraph and the previous one must be included in Discussion section since adding relevant information not initially taken into account. Authors should present an approximate prevalence of HCV infection in their population based on their data. With this they could really estimate an impact of HCV patients among Methadone users and a potential improvement by applying eradicating therapies. Linking with previous: Based on DAA potential costs, estimated prevalence and the costs from complications of advanced HCV patients (cirrhotic), authors could attempt to explore a costs-effects analysis (Markov model or similar) to propose intervention strategies based on their local data and resources. Introduction section must be shortened. Discussion should be shortened/modified according to my previous suggestions.

Response: Thanks for the reviewer’s suggestion. We think the results of our study may be applied to some developed countries where DAA regimens are exclusively used. First, although the DAAs could removes PR-related barriers (i.e., low SVR, numerous contraindications), they may not fully eliminate all barriers the patients in MMT faced (i.e., high cost, mild symptoms). Second, we also collected patients’ willingness and barriers to accepting DAAs, the results can be adopted in the developed countries to facilitate access to treatment.

Introduction and Discussion section had been modified according to the reviewer’s suggestion. We also added the prevalence of HCV infection in our population (See line 185 and line 266). We appreciated the reviewer’s comments that it would be very interesting to know the best candidates for therapies in resource-limited settings, and we have added a sentence to state “Further analyses are also needed to determine the cost-effectiveness of early treatment for chronic HCV patients in China and other resource-limited settings.”(See line 315-316).

Round 2

Reviewer 1 Report

I am happy to see the authors addressed many of my main concerns and strengthened the manuscript's conclusions. I will recommend the publication of this version. 

Reviewer 2 Report

As expected for a revised version sent shortly after my initial review (10 days!!!), only a few minor and irrelevant changes are provided by authors. Essentially, their responses to suggestions are poor, when existing. The major points suggested in my review have been totally obviated: cost-effectiveness analysis (which could be both done based on IFN and/or DAA costs), HCV fibrosis stage, length for Introduction and Discussion sections, etc.  

My opinion regarding an article does not change in a temporal manner. Besides the mostly unmodified revised manuscript presented, I think that Authors have not even done the effort to answer in a rational and convincent way my suggestions/recommendations.